# Numerically Solving Parametric Families of High-Dimensional Kolmogorov Partial Differential Equations via Deep Learning

**Julius Berner**[*]
Faculty of Mathematics, University of Vienna
Oskar-Morgenstern-Platz 1, 1090 Vienna, Austria
julius.berner@univie.ac.at

**Markus Dablander**[*]
Mathematical Institute, University of Oxford
Andrew Wiles Building, OX2 6GG, Oxford, United Kingdom
markus.dablander@maths.ox.ac.uk

**Philipp Grohs**
Faculty of Mathematics and Research Platform DataScience@UniVienna, University of Vienna
Oskar-Morgenstern-Platz 1, 1090 Vienna, Austria
RICAM, Austrian Academy of Sciences
Altenberger Straße 69, 4040 Linz, Austria
philipp.grohs@univie.ac.at

## Abstract

We present a deep learning algorithm for the numerical solution of parametric families of high-dimensional linear Kolmogorov partial differential equations (PDEs). Our method is based on reformulating the numerical approximation of a whole family of Kolmogorov PDEs as a single statistical learning problem using the Feynman-Kac formula. Successful numerical experiments are presented, which empirically confirm the functionality and efficiency of our proposed algorithm in the case of heat equations and Black-Scholes option pricing models parametrized by affine-linear coefficient functions. We show that a single deep neural network trained on simulated data is capable of learning the solution functions of an entire family of PDEs on a full space-time region. Most notably, our numerical observations and theoretical results also demonstrate that the proposed method does not suffer from the curse of dimensionality, distinguishing it from almost all standard numerical methods for PDEs.

## 1 Introduction

Linear parabolic partial differential equations (PDEs) of the form

$$\frac{\partial u_\gamma}{\partial t} = \frac{1}{2}\operatorname{Trace}\left(\sigma_\gamma[\sigma_\gamma]^* \nabla_x^2 u_\gamma\right) + \langle \mu_\gamma, \nabla_x u_\gamma \rangle, \quad u_\gamma(x,0) = \varphi_\gamma(x), \tag{1}$$

are referred to as Kolmogorov PDEs, see [23] for a thorough study of their mathematical properties. Throughout this paper, the functions

$$\varphi_\gamma : \mathbb{R}^d \to \mathbb{R} \quad \text{(initial condition)} \quad \text{and} \quad \sigma_\gamma : \mathbb{R}^d \to \mathbb{R}^{d\times d}, \quad \mu_\gamma : \mathbb{R}^d \to \mathbb{R}^d \quad \text{(coefficient maps)}$$

---
[*]Equal contribution.

are continuous, and are implicitly determined by a real parameter vector $\gamma \in D$, whereby $D$ is a compact set in Euclidean space.

Kolmogorov PDEs frequently appear in applications in a broad variety of scientific disciplines, including physics and financial engineering [9, 42, 58]. In particular, note that the heat equation from physical modelling as well as the widely-known Black-Scholes equation from computational finance are important special cases of Equation (1). Typically, one is interested in finding the (viscosity) solution[2]

$$u_\gamma : [v, w]^d \times [0, T] \to \mathbb{R}$$

of Equation (1) on a predefined space-time region of the form $[v, w]^d \times [0, T]$. In almost all cases, however, Kolmogorov PDEs cannot be solved explicitly. Furthermore, standard numerical solution algorithms for PDEs, in particular those based on a discretization of the considered domain, are known to suffer from the so-called *curse of dimensionality*[3], meaning that their computational cost grows exponentially in the dimension of the domain [2, 51]. The development of new, computationally efficient methods for the numerical solution of Kolmogorov PDEs is therefore of high interest for applied scientists.

In this work, we present a novel deep learning algorithm capable of numerically approximating the solutions $(u_\gamma)_{\gamma \in D}$ of a whole family of $\gamma$-parametrized Kolmogorov PDEs on a full space-time region. Specifically, our proposed method allows to train a single deep neural network

$$\Phi \colon D \times [v, w]^d \times [0, T] \to \mathbb{R} \tag{2}$$

to approximate the *parametric solution map*

$$\bar{u} : D \times [v, w]^d \times [0, T] \to \mathbb{R}, \quad (\gamma, x, t) \mapsto \bar{u}(\gamma, x, t) := u_\gamma(x, t), \tag{3}$$

of a family of $\gamma$-parametrized Kolmogorov PDEs on the generalized domain $D \times [v, w]^d \times [0, T]$. Most notably, we also theoretically investigate the associated approximation and generalization errors and rigorously show that our algorithm does not suffer from the curse of dimensionality with respect to the neural network size as well as the sample size. We emphasize that our empirical results strongly suggest that also the empirical risk minimization (ERM) algorithm, usually a variant of stochastic gradient descent, does not suffer from the curse of dimensionality but proving this is out of scope of this paper.

## 1.1   PDEs and Deep Learning: Current Research and Related Work

Interest in deep-learning based techniques for the numerical solution of PDEs has been growing rapidly in recent years [6, 24, 32, 47, 53, 56, 57]. This sharp rise in interest can partly be explained by the remarkable ability of deep neural networks to avoid incurring the curse of dimensionality when used to approximate PDE solutions in particular settings. More specifically, in some situations it has been possible to find theoretical upper bounds for the size of the required neural network architectures which do not depend exponentially on the dimension of the PDE [14, 20, 29, 50, 48]. This represents a rare and crucial advantage over classical finite difference and finite element methods, all of which typically cannot be used in high dimensions due to the resulting exponential explosion of required computational costs.

As a result of these successes, deep learning has recently been studied as a numerical solution technique for the more general group of parametric PDEs and their associated parametric solution maps [12, 27, 33, 36, 37, 50]. The investigation of the capabilities of deep artificial neural networks to learn parametric solution maps of specific parametrizable families of PDEs has become a new and active area of research. In this work, we provide novel theoretical and empirical results which, for the first time, demonstrate the viability of deep learning algorithms for the scalable solution of large classes of parametric Kolmogorov PDEs.

The formulation of the learning problem underlying our method is inspired by the work of Beck et al. [5]. There it is shown how deep neural networks can be used to numerically solve a non-parametric version of Equation (1) with fixed initial condition $\varphi_\gamma$ and fixed coefficients maps $\sigma_\gamma, \mu_\gamma$

on a predefined space region $[v, w]^d$ and at a predefined time slice $T > 0$. In other words, their non-parametric method allows to approximate the function

$$u_\gamma( \cdot, T) : [v, w]^d \to \mathbb{R}, \quad x \mapsto u_\gamma(x, T),$$

for fixed $\gamma \in D$ by training a deep neural network with suitable simulated data of the form

$$(X, \varphi_\gamma(S_{\gamma, X, T})) \in [v, w]^d \times \mathbb{R}.$$

Here, $X$ is uniformly drawn from the spatial hypercube $[v, w]^d$ and the random vector $S_{\gamma, X, T}$ is the value of the solution process $(S_{\gamma, X, t})_{t \geq 0}$ of the stochastic differential equation (SDE)

$$dS_{\gamma, X, t} = \mu_\gamma(S_{\gamma, X, t})dt + \sigma_\gamma(S_{\gamma, X, t})dB_t, \quad S_{\gamma, X, 0} = X,$$

at time $t = T$, whereby $(B_t)_{t \geq 0}$ is a standard $d$-dimensional Brownian motion.

The choice of training data is based on the following important identity, which under suitable regularity assumptions holds for all $x \in [v, w]^d$, $t \in [0, T]$, and $\gamma \in D$:

$$u_\gamma(x, t) = \mathbb{E}[\varphi_\gamma(S_{\gamma, x, t})]. \tag{4}$$

Equality (4) is a version of the well-known *Feynman-Kac formula* which establishes a link between the theory of parabolic PDEs and the theory of stochastic differential equations [23]. Using the Feynman-Kac formula, one can show within the mathematical framework of empirical risk minimization [11, 55] that $u_\gamma(\cdot, T)$ is in fact the solution of the supervised statistical learning problem defined by the predictor variable $X$, the target variable $\varphi_\gamma(S_{\gamma, X, T})$, and a standard quadratic loss function [5].

## 1.2 Novel Contribution

In this work, we introduce the first algorithm for the numerical solution of *parametric* Kolmogorov PDEs on a *whole* space-time region. No previous technique has achieved this degree of generality; all former methods for parametric Kolmogorov PDEs were either only capable of computing local solutions at single space-time points of the domain using Monte Carlo based approaches or did not employ deep neural networks and were thus not able to break the curse of dimensionality. Our technique is made possible by constructing a suitable supervised learning problem via a nontrivial application of the Feynman-Kac formula (4), which involves random initial conditions and SDEs with random coefficients and stopping times. This reformulation of a broad class of parametric PDEs as learning problems provides a new theoretical framework to analyze the convergence behavior of deep learning algorithms. Building upon this framework, we prove theoretical guarantees for the computational performance of our technique and, to the best of our knowledge, establish the first combined approximation and generalization results for parametric PDEs.

Note that the parametric nature of the presented algorithm opens up the novel possibility to study changes in the potentially high-dimensional solution manifold of Equation (1) as its initial conditions and coefficient maps vary with $\gamma \in D$. The study of such changes is commonly referred to as *sensitivity analysis*. Recall that the proposed method delivers a neural network $\Phi$ which approximates the parametric PDE solution map, i.e. $\Phi \approx \bar{u}$. The partial derivatives of $\Phi$ with respect to the parameter $\gamma$, the spatial variable $x$, and the time variable $t$ can then be readily computed via automatic differentiation. Thus, the partial derivatives of $\Phi$ become computationally accessible approximations of the partial derivatives of $\bar{u}$. The partial derivatives of $\bar{u}$ in turn play an important role in a variety of widespread applications, such as in the computation of the "Greeks" associated with the Black-Scholes model from financial engineering (see Section 3.1).

Another highly relevant application area opened up by our method is the calibration of the usually unknown PDE coefficients $\sigma_\gamma, \mu_\gamma$ using real-world data. After solving a parametric PDE with our technique, one can fit $\gamma$ such that the PDE solution manifold best describes a real data set and additionally apply uncertainty quantification techniques if $\gamma$ is modelled as a random variable.

Finally, we establish a new architecture and compare different learning schemes to provide future researchers with a robust framework for parametric PDEs, which are inherently less stable than their simpler non-parametric counterparts. Further, this work is complemented by an extendable implementation with the possibility of distributed training and hyperparameter optimization for the special use-cases of other researchers.

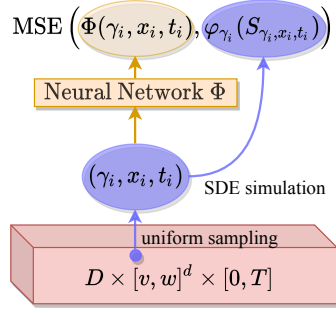

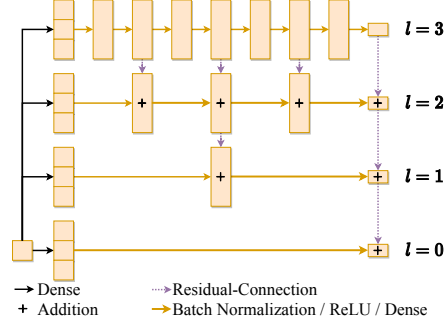

Figure 1: Illustration of the proposed supervised learning problem with predictor variable $\Lambda$ and target variable $\varphi_\Gamma(S_{\Gamma,X,\mathcal{T}})$.

Figure 2: Illustration of the Multilevel architecture for $L = 4$, $q = 3$.

## 2   Algorithm

The key idea of the presented algorithm is to describe the parametric solution map (3) of the $\gamma$-parametrized Kolmogorov PDE (1) as the regression function of an appropriately chosen supervised statistical learning problem. One can then use simulated training data in order to learn $\bar{u}$ by means of deep learning. Inspired by the Feynman-Kac formula (4), we construct a new statistical learning problem via a uniformly distributed predictor variable and a statistically dependent target variable:

$$\Lambda := (\Gamma, X, \mathcal{T}) \in D \times [v,w]^d \times [0,T] \quad \text{(predictor)} \quad \text{and} \quad Y := \varphi_\Gamma(S_\Lambda) \in \mathbb{R} \quad \text{(target)}.$$

The random vector $S_\Lambda$ is defined as the value of the solution process $(S_{\Gamma,X,t})_{t\geq 0}$ of the $\Gamma$-parametrized stochastic differential equation

$$dS_{\Gamma,X,t} = \mu_\Gamma(S_{\Gamma,X,t})dt + \sigma_\Gamma(S_{\Gamma,X,t})dB_t, \quad S_{\Gamma,X,0} = X, \tag{5}$$

at the (random) stopping time $t = \mathcal{T}$. For suitable regularity assumptions, the Feynman-Kac formula (4) then ensures that

$$\mathbb{E}[Y \mid \Lambda = (\gamma, x, t)] = \mathbb{E}[\varphi_\Gamma(S_\Lambda) \mid \Lambda = (\gamma, x, t)] = \mathbb{E}[\varphi_\gamma(S_{\gamma,x,t})] = u_\gamma(x,t) = \bar{u}(\gamma, x, t).$$

This shows that the minimizer of the corresponding statistical learning problem with quadratic loss function is indeed the parametric Kolmogorov PDE solution map, see Section A.1 in the appendix for the precise assumptions and a detailed proof.

**Theorem 1** (Learning Problem). *It holds that the parametric solution map $\bar{u}$ is the unique minimizer of the statistical learning problem*

$$\min_f \mathbb{E}\big[\big(f(\Lambda) - Y\big)^2\big]. \tag{6}$$

Restricting to a hypothesis space of suitable neural networks $\mathcal{H}$ and minimizing the empirical mean squared error (MSE) loss corresponding to (6), we arrive at the feasible supervised ERM problem

$$\min_{\Phi \in \mathcal{H}} \tfrac{1}{s} \sum_{i=1}^s (\Phi(\lambda_i) - y_i)^2 \tag{7}$$

where $((\lambda_i, y_i))_{i=1}^s$ are realizations of i.i.d. samples drawn from the distribution of $(\Lambda, Y)$. Typically, this problem is then solved by a variant of stochastic gradient descent [49]. The algorithm is graphically illustrated in Figure 1.

It is trivial to simulate i.i.d. samples of the predictor variable $\Lambda$, due to its uniform distribution. On the other hand i.i.d. samples of the target variable $Y = \varphi_\Gamma(S_\Lambda)$ can be obtained via standard numerical SDE solution techniques without curse of dimensionality [35]. An example for such a technique is given by the *Euler-Maruyama approximation* with $M \in \mathbb{N}$ equidistant steps $(S_\Lambda^{M,m})_{m=0}^M$ which is defined by the following scheme:

$$S_\Lambda^{M,0} = X \quad \text{and} \quad S_\Lambda^{M,m+1} = S_\Lambda^{M,m} + \mu_\Gamma(S_\Lambda^{M,m})\tfrac{\mathcal{T}}{M} + \sigma_\Gamma(S_\Lambda^{M,m})\big(B_{\frac{(m+1)\mathcal{T}}{M}} - B_{\frac{m\mathcal{T}}{M}}\big). \tag{8}$$

The following theorem shows that solving the learning problem with data simulated by the Euler-Maruyama scheme does indeed result in the expected approximation of the parametric PDE solution map $\bar{u}$, see Section A.1 in the appendix for a proof.

**Theorem 2** (Approximated Learning Problem). *The unique minimizer $\bar{u}^M$ of the approximated statistical learning problem*

$$\min_f \mathbb{E}\big[\big(f(\Lambda) - Y^M\big)^2\big]$$

*where $Y^M := \varphi_\Gamma(S_\Lambda^{M,M})$ is simulated using the Euler-Maruyama scheme* (8) *with $M \sim 1/\varepsilon^2$ equidistant steps satisfies that*

$$\|\bar{u}^M - \bar{u}\|_{\mathcal{L}^\infty(D \times [v,w]^d \times [0,T])} \leq \varepsilon.$$

In other words, the approximation of the SDE solution $S_\Lambda^{M,M} \approx S_\Lambda$ carries over to the approximation of the corresponding minimizer $\bar{u}^M \approx \bar{u}$. Therefore there exists no constraint of having to solve the SDE in (5) analytically. The ability to easily simulate artificial training data opens up the highly desirable capability to supply the learning algorithm with a potentially infinite stream of i.i.d. data samples. Instead of having to use a train/val/test split on a given finite data set, one can thus constantly simulate new data points on demand during training. Since the number of samples then grows at will parallel to the training process, the first epoch never finishes and every new gradient computation can be done on the basis of previously unseen data.

## 2.1 Example: Affine-Linear Coefficient Functions

In Section 3 we present numerical experiments based on our algorithm for the important special case where $\sigma_\gamma$ and $\mu_\gamma$ are affine-linear functions. Thus from now on let us assume that[4]

$$\sigma_\gamma : \mathbb{R}^d \to \mathbb{R}^{d \times d}, \quad \sigma_\gamma(x) = [\gamma_{\sigma,1}x| \ldots |\gamma_{\sigma,d}x] + \gamma_{\sigma,d+1},$$
$$\mu_\gamma : \mathbb{R}^d \to \mathbb{R}^d, \quad \mu_\gamma(x) = \gamma_{\mu,1}x + \gamma_{\mu,2},$$

are affine-linear functions, which are determined by parameter tuples of matrices and vectors

$$\gamma_\sigma \in D_\sigma \subseteq (\mathbb{R}^{d \times d})^{d+1}, \quad \gamma_\mu \in D_\mu \subseteq \mathbb{R}^{d \times d} \times \mathbb{R}^d.$$

The parameter sets $D_\sigma$ and $D_\mu$ are chosen to be compact. Together with a suitable compact parameter set $D_\varphi \subseteq \mathbb{R}^k$ for the initial function $\varphi_\gamma$, we obtain

$$\gamma := (\gamma_\sigma, \gamma_\mu, \gamma_\varphi) \in D_\sigma \times D_\mu \times D_\varphi := D \subseteq (\mathbb{R}^{d \times d})^{d+1} \times (\mathbb{R}^{d \times d} \times \mathbb{R}^d) \times \mathbb{R}^k.$$

This leads to an input dimension of our neural network $\Phi$ of

$$\dim_{\text{in}}(\Phi) = \dim(D \times [v,w]^d \times [0,T]) = d^3 + 2d^2 + 2d + 1 + k.$$

Kolmogorov PDEs with affine-linear coefficient functions regularly appear in applications; the heat equation from physics and the classical and generalized Black-Scholes equations from computational finance are important examples of Kolmogorov PDEs with affine-linear coefficient maps [13, 45]. Note that while affine-linear coefficient functions are important in practise, computationally fast to evaluate, and easy to parametrize, the presented method is not restricted to the case of affine-linear coefficients and can as well be used in a substantially more general setting.

## 3 Numerical Results

We implemented the framework described in Section 2 in PyTorch [43] and computed our results on a Nvidia DGX-1 using Tune [39] for experiment execution and hyperparameter optimization. In this section, we describe our setting and present four encouraging demonstrations of the performance of our algorithm.[5]

For the neural network $\Phi$ we propose a *Multilevel architecture* which is inspired by multilevel techniques such as Multilevel Monte Carlo methods [18], network architectures in [26, 61], and the architecture for the squaring function used in the proofs of our theoretical results in Section A.1, see also [60, Figure 2c]. One can view the output of the network $\Phi$ as a sum $\sum_{l=0}^{L-1} \Phi_l$ of sub-networks

$\Phi_l$ with $2^l$ layer each. We think of the shallow network $\Phi_0$ as computing a coarse approximation of $\bar{u}$ and of the deep networks $\Phi_l$ (with $l \geq 1$) as approximately learning the residuals $\bar{u} - \sum_{i=0}^{l-1} \Phi_i$. To facilitate optimization we normalize our inputs[6] and to enhance expressivity we add an initial layer which increases the width by a given factor $q$. The Multilevel architecture is depicted in Figure 2 and described in detail in Definition 1 in the appendix.

Our optimized hyperparameters as well as an ablation study of our architecture and training scheme can be found in Sections A.2 and A.3 in the appendix. For all our experiments we were able to stick to a similar setup which depicts its robustness and general applicability. This is also mirrored by the small standard deviations of our considered errors across independent runs in Tables 1, 2, 3, and 4. These tables report average runtimes (in seconds), average approximation errors, and their standard deviations w.r.t. 4 independent runs each 4000 gradient descent steps. As an evaluation metric we approximately computed $\mathcal{L}^1$-errors via Monte Carlo sampling, that is

$$\left\| \frac{\Phi(\Lambda) - \bar{u}(\Lambda)}{1 + |\bar{u}(\Lambda)|} \right\|_{\mathcal{L}^1} := \mathbb{E}\left[ \frac{|\Phi(\Lambda) - \bar{u}(\Lambda)|}{1 + |\bar{u}(\Lambda)|} \right] \approx \frac{1}{n} \sum_{i=1}^{n} \left( \frac{|\Phi(\lambda_i) - \bar{u}(\lambda_i)|}{1 + |\bar{u}(\lambda_i)|} \right) \tag{9}$$

with $n \in \mathbb{N}$ realizations $(\lambda_i)_{i=1}^{n}$ of i.i.d. samples drawn from the distribution of $\Lambda$ (drawn independently of the training data in (7) and drawn independently for each evaluation step). This means that we always evaluate our model w.r.t. to the parametric solution map $\bar{u}$ on unseen input data; if no closed-form solution for $\bar{u}$ is available, as in the case of the Basket option in 3.2 below, we approximate $\bar{u}(\lambda_i)$ pointwise via Monte Carlo sampling, i.e.

$$\bar{u}(\lambda_i) = \bar{u}(\gamma_i, x_i, t_i) = \mathbb{E}[\varphi_{\gamma_i}(S_{\gamma_i, x_i, t_i})] \approx \frac{1}{m} \sum_{j=1}^{m} \varphi_{\gamma_i}(s_j) \tag{10}$$

where $(s_j)_{j=1}^{m}$ are realizations of i.i.d. samples drawn from the distribution of the Euler-Maruyama approximation $S_{\lambda_i}^{M,M}$ (drawn independently of the training data in (7) and drawn independently for each point and each evaluation step). Note that (9) is invariant under scaling of the hypercubes and locally corresponds to relative errors where the solution $\bar{u}$ is large and absolute errors where it is small, which in particular prevents division by zero.

## 3.1 Black-Scholes Options Pricing Model

Our first example shows that neural networks are capable of learning a parametric version of the highly-celebrated Black-Scholes option pricing model [9]. We consider a European put option which gives its owner the right, but not the obligation, to sell a single underlying financial asset at a specified strike price and at a given time. Formally, this corresponds to $d = 1$ and

$$\sigma_\gamma(x) = \gamma_\sigma x, \quad \mu_\gamma(x) = 0, \quad \varphi_\gamma(x) = \max\{\gamma_\varphi - x, 0\}, \quad x \in \mathbb{R},$$

with[7] $\gamma_\sigma \in D_\sigma \subseteq \mathbb{R}$ and $\gamma_\varphi \in D_\varphi \subseteq \mathbb{R}$. Effectively, this leads to an input dimension of our neural network $\Phi$ of $\dim_{\text{in}}(\Phi) = 4$. In case of the present Black-Scholes model, the associated SDE in (5) can actually be solved explicitly; it gives rise to geometric Brownian motion with uniformly distributed volatility $\Gamma_\sigma \in D_\sigma$ and stopping time $\mathcal{T} \in [0, T]$, i.e.

$$S_\Lambda = X e^{-0.5 \mathcal{T} \Gamma_\sigma^2 + \sqrt{\mathcal{T}} \Gamma_\sigma N}, \quad N \sim \mathcal{N}(0, 1).$$

We thus obtain an analytic expression for the parametric PDE solution,

$$\bar{u}(\gamma, x, t) = \gamma_\varphi \Psi(h_\gamma(x, t) + \sqrt{t}\, \gamma_\sigma) - x \Psi(h_\gamma(x, t)),$$

and the partial derivatives, e.g.

$$\frac{\partial \bar{u}}{\partial \gamma_\sigma}(\gamma, x, t) = x \sqrt{t} \Psi'(-h_\gamma(x, t)),$$

where

$$\Psi(z) := \frac{1}{2}\left[ 1 + \text{erf}\left( \frac{z}{\sqrt{2}} \right) \right] \quad \text{and} \quad h_\gamma(x, t) := -\frac{1}{\sqrt{t}\, \gamma_\sigma}\left[ \ln\left( \frac{x}{\gamma_\varphi} \right) + \frac{t \gamma_\sigma^2}{2} \right],$$

see [4, Section 13.7]. This analytic expression can be used to evaluate the performance of our algorithm. We point out that the partial derivatives of $\bar{u}$ are crucial in option pricing and each of

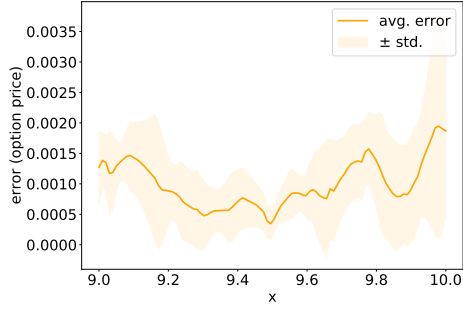
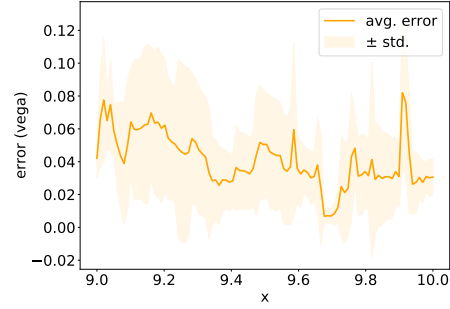

Figure 3: Shows the average prediction error $\frac{|\Phi(\gamma,\cdot,t)-\bar{u}(\gamma,\cdot,t)|}{1+|\bar{u}(\gamma,\cdot,t)|}$ and its standard deviation at $t=0.5$, $\gamma_\sigma = 0.35$, and $\gamma_\varphi = 11$.

Figure 4: Shows the average error of the Vega $\frac{|\frac{\partial \Phi}{\partial \gamma_\sigma}(\gamma,\cdot,t)-\frac{\partial \bar{u}}{\partial \gamma_\sigma}(\gamma,\cdot,t)|}{1+|\frac{\partial \bar{u}}{\partial \gamma_\sigma}(\gamma,\cdot,t)|}$ and its standard deviation at $t=0.5$, $\gamma_\sigma = 0.35$, and $\gamma_\varphi = 11$.

them is associated with a distinct economic interpretation. They are often referred to as *Greeks* and describe the sensitivity of the option price w.r.t. different model parameters, see for instance [4, 52]. The most prominent Greeks are given by

$$\Delta = \frac{\partial \bar{u}}{\partial x}, \quad \text{Vega} = \frac{\partial \bar{u}}{\partial \gamma_\sigma}, \quad \Theta = -\frac{\partial \bar{u}}{\partial t}.$$

On the basis of the proposed algorithm, our neural network $\Phi$ is capable of learning the parametric solution map $\bar{u}$ of the above problem in $24000$ gradient updates up to an average $\mathcal{L}^1$-error of $0.0011$, see Table 1 and Figure 3. As expected, the partial derivatives of our network $\Phi$ (computed via automatic differentiation) approximate the partial derivatives of $\bar{u}$ as can be seen in Figure 4. Further evidence can be found in Figures 5, 6, 7, and 8 in the appendix. Even though the parametric PDE problem can be solved explicitly in this special case, we use this relatively simple example for the purpose of illustrating our algorithm in an intuitive setting. As we will see below, the proposed algorithm is by no means restricted to such basic examples and can be applied successfully to much more complex and high-dimensional problems as well.

## 3.2 Basket Put Option

In the following we show that we can obtain comparable results to Section 3.1 in the case of a considerably more complicated Basket put option pricing problem, where analytical solutions of the PDE and the SDE are lacking. In such cases, our algorithm allows practitioners to nevertheless gain valuable insights into the behaviour of the PDE solution manifold as input parameters vary. By means of our trained model $\Phi$ one can easily compute sensitivity values $\frac{\partial \Phi}{\partial \gamma} \approx \frac{\partial \bar{u}}{\partial \gamma}$, $\frac{\partial \Phi}{\partial t} \approx \frac{\partial \bar{u}}{\partial t}$, and $\frac{\partial \Phi}{\partial x} \approx \frac{\partial \bar{u}}{\partial x}$ via automatic differentiation or fit the parameter $\gamma$ to a real-world data-set $((x_i, t_i), u_\gamma(x_i, t_i))_{i=1}^m$ with unknown $\gamma$ by minimizing $\min_{\gamma \in D} \sum_{i=1}^m \left( \Phi(\gamma, x_i, t_i) - u_\gamma(x_i, t_i) \right)^2$ via stochastic gradient descent. Moreover, one can obtain estimates for probabilistic quantities related to uncertainty such as

$$\mathbb{V}[u_\Xi(x,t)] \approx \mathbb{V}[\Phi(\Xi, x, t)] \approx \frac{1}{m-1} \sum_{i=1}^m \left( \Phi(\xi_i, x, t) - \frac{1}{n} \sum_{j=1}^m \Phi(\xi_j, x, t) \right)^2$$

where $(\xi_i)_{i=1}^m$ are realizations of i.i.d. samples drawn from the distribution of a random variable $\Xi$ of interest. None of these types of insights were accessible before the presented deep learning method.

We proceed by demonstrating the performance of the proposed algorithm for a general multidimensional affine-linear setting as described in Section 2.1. To this end, let $d = 3$ and define the initial condition via

$$\varphi_\gamma(x) = \max\left\{ \gamma_\varphi - \tfrac{1}{3} \sum_{i=1}^3 x_i, 0 \right\}, \quad x \in \mathbb{R}^3,$$

with $\gamma_\varphi \in D_\varphi \subseteq \mathbb{R}$. This corresponds to the situation of a Basket put option in a very general multidimensional Black-Scholes model with 3 potentially highly correlated assets. Note that within the above setup, the input dimension of our neural network $\Phi$ is given by

$$\dim_{\text{in}}(\Phi) = d^3 + 2d^2 + 2d + 1 + 1 = 53.$$

To generate samples of our target variable $\varphi_\Gamma(S_\Lambda)$, we simulate solutions of the SDE in (5) using the Euler-Maruyama scheme (8) with $M = 25$ equidistant steps. Moreover, we use a Monte Carlo

| Table 1: Results for the Black-Scholes model | | |
|---|---|---|
| step | avg. time (s) | avg. $\mathcal{L}^1$-error |
| 0 | $0 \pm 0$ | $0.6812 \pm 0.0704$ |
| 4k | $471 \pm 3$ | $0.0088 \pm 0.0056$ |
| 8k | $943 \pm 6$ | $0.0062 \pm 0.0025$ |
| 12k | $1413 \pm 9$ | $0.0030 \pm 0.0004$ |
| 16k | $1885 \pm 11$ | $0.0017 \pm 0.0001$ |
| 20k | $2356 \pm 14$ | $0.0013 \pm 0.0002$ |
| 24k | $2827 \pm 17$ | $0.0011 \pm 0.0001$ |

| Table 2: Results for the Basket option | | |
|---|---|---|
| step | avg. time (s) | avg. $\mathcal{L}^1$-error |
| 0 | $0 \pm 0$ | $0.7912 \pm 0.0276$ |
| 4k | $811 \pm 7$ | $0.0131 \pm 0.0019$ |
| 8k | $1614 \pm 4$ | $0.0087 \pm 0.0013$ |
| 12k | $2434 \pm 28$ | $0.0062 \pm 0.0009$ |
| 16k | $3236 \pm 27$ | $0.0058 \pm 0.0011$ |
| 20k | $4162 \pm 154$ | $0.0046 \pm 0.0007$ |
| 24k | $5077 \pm 291$ | $0.0042 \pm 0.0002$ |
| 28k | $6024 \pm 463$ | $0.0039 \pm 0.0001$ |

approximation with $m = 2^{20}$ samples to compute the pointwise evaluation of the reference solution $\bar{u}(\lambda_i)$ according to (10) as needed for the error estimation in (9). Despite the considerably higher dimension of this problem compared with the previous problem from Section 3.1, our deep learning approach shows almost the same approximation behavior, see Table 2. This remarkably weak dependence on the dimension of the input data is further supported by the next examples from physical modelling, where we shall increase the dimensionality of the studied problems even further.

## 3.3 Heat Equation with Varying Diffusion Coefficients

In this Section, we present two examples of high-dimensional heat equations in $d = 10$ and $d = 150$ dimensions with paraboloid and Gaussian initial conditions

$$\varphi_\gamma(x) = \|x\|^2 \quad \text{(paraboloid)} \quad \text{and} \quad \varphi_\gamma = e^{-\|x\|^2} \quad \text{(Gaussian)}.$$

This formally corresponds to

$$\sigma_\gamma(x) = \gamma_\sigma \quad \text{and} \quad \mu_\gamma(x) = 0$$

where we use a matrix $\gamma_\sigma \in D_\sigma \subseteq \mathbb{R}^{10 \times 10}$ for the paraboloid case and a scalar $\gamma_\sigma \in D_\sigma \subseteq \mathbb{R}$ in the Gaussian case, leading to input dimensions of our models $\Phi$ of

$$\dim_{\text{in}}(\Phi) = d^2 + d + 1 = 111 \quad \text{(paraboloid)} \quad \text{and} \quad \dim_{\text{in}}(\Phi) = d + 1 + 1 = 152 \quad \text{(Gaussian)}.$$

Notice that here the solution of the corresponding SDE can be directly sampled via a Brownian motion with random initial position $X$ and stopping time $\mathcal{T}$, i.e.

$$S_\Lambda = X + \sqrt{\mathcal{T}} \Gamma_\sigma N, \quad N \sim \mathcal{N}(0, I_d),$$

see [5, Section 3.2]. For evaluation purposes, these examples were purposefully constructed to have analytic expressions for the parametric solution maps $\bar{u}$, which are given by

$$\bar{u}(\gamma_\sigma, x, t) = \|x\|^2 + t \operatorname{Trace}(\gamma_\sigma \gamma_\sigma^*) \quad \text{(paraboloid)}, \quad \bar{u}(\gamma_\sigma, x, t) = \frac{e^{-\frac{\|x\|^2}{1 + 2t\gamma_\sigma^2}}}{(1 + 2t\gamma_\sigma^2)^{d/2}} \quad \text{(Gaussian)}.$$

However, in almost all other practical cases an analytic solution for $\bar{u}$ is impossible to obtain and numerical methods are the only path forward.

The above dimensionality settings represent regimes which are completely out of scope for all preexisting numerical schemes. Nevertheless, Tables 3 and 4 confirm that our proposed deep learning method once again efficiently converges to the desired parametric solution map $\bar{u}$. Our results empirically demonstrate that, contrary to conventional numerical solvers, our deep learning based method does not suffer from the curse of dimensionality, see also Figure 9 in the appendix. We will rigorously prove this fact in the next section.

## 4 Theoretical Guarantees

As a first example, we stick to the heat equation with paraboloid initial condition from above and show that neural networks are capable of simultaneously approximating the parametric solution map $\bar{u}$ and its gradient with the number of network parameters scaling only polynomially in the dimension $d$. Such an approximation guarantee without curse of dimensionality ensures that sensitivity analysis is possible even in very high dimensions. A proof can be found in Section A.1 in the appendix.

Table 3: Results for the heat equation with paraboloid initial condition

| step | avg. time (s) | avg. $\mathcal{L}^1$-error |
|------|---------------|----------------------------|
| 0 | $0 \pm 0$ | $0.9609 \pm 0.0052$ |
| 4k | $1904 \pm 19$ | $0.0150 \pm 0.0008$ |
| 8k | $3808 \pm 37$ | $0.0120 \pm 0.0007$ |
| 12k | $5712 \pm 57$ | $0.0093 \pm 0.0006$ |
| 16k | $7616 \pm 76$ | $0.0068 \pm 0.0001$ |
| 20k | $9520 \pm 95$ | $0.0062 \pm 0.0003$ |
| 24k | $11424 \pm 114$ | $0.0057 \pm 0.0001$ |
| 28k | $13328 \pm 132$ | $0.0056 \pm 0.0000$ |

Table 4: Results for the heat equation with Gaussian initial condition

| step | avg. time (s) | avg. $\mathcal{L}^1$-error |
|------|---------------|----------------------------|
| 0 | $0 \pm 0$ | $0.2035 \pm 0.0714$ |
| 4k | $2070 \pm 40$ | $0.0123 \pm 0.0047$ |
| 8k | $4131 \pm 82$ | $0.0050 \pm 0.0018$ |
| 12k | $6192 \pm 124$ | $0.0051 \pm 0.0022$ |
| 16k | $8258 \pm 165$ | $0.0033 \pm 0.0015$ |
| 20k | $10323 \pm 206$ | $0.0025 \pm 0.0011$ |
| 24k | $12388 \pm 247$ | $0.0024 \pm 0.0008$ |
| 28k | $14454 \pm 290$ | $0.0019 \pm 0.0002$ |

**Theorem 3** (Sobolev Approximation). *There exists a neural network $\Phi$ with ReLU activation function and $\mathcal{O}(d^4 \log(d/\varepsilon))$ parameters satisfying that*

$$\|\Phi - \bar{u}\|_{\mathcal{L}^\infty(D \times [v,w]^d \times [0,T])} \leq \varepsilon \quad and \quad \|\nabla\Phi - \nabla\bar{u}\|_{\mathcal{L}^\infty(D \times [v,w]^d \times [0,T])} \leq \varepsilon.$$

Let us now consider the heat equation with varying diffusivity and Gaussian initial condition. In fact, our framework allows us to rigorously prove sample complexity estimates for this problem which represents an almost unique scenario for deep learning based methods. This is rendered possible by the structure of the underlying parametric Kolmogorov PDE and its associated SDE which allows us to describe the distribution of the predictor and target variable, simulate i.i.d. samples, and infer regularity properties on the regression function. We briefly sketch the theorem in the following; the precise formulation and the proof can be found in the appendix.

**Theorem 4** (Generalization). *Using $s \sim (d/\varepsilon)^2 \operatorname{polylog}(d/\varepsilon)$ many samples, every empirical risk minimizer $\hat{\Phi}$ of (7) in a suitable hypothesis space $\mathcal{H}$ of neural networks with ReLU activation function, $\mathcal{O}(\operatorname{polylog}(d/\varepsilon))$ layers, $\mathcal{O}(d)$ neurons per layer, and parameters bounded by $\mathcal{O}(1)$ satisfies with high probability that*

$$\tfrac{1}{V}\|\hat{\Phi} - \bar{u}\|_{\mathcal{L}^2(D \times [v,w]^d \times [0,T])}^2 \leq \varepsilon$$

*where $V := \operatorname{vol}(D \times [v,w]^d \times [0,T])$.*

Note that it holds that

$$\tfrac{1}{V}\| \cdot \|_{\mathcal{L}^2(D \times [v,w]^d \times [0,T])}^2 = \| \cdot \|_{\mathcal{L}^2(\mathbb{P}_\Lambda)}^2$$

where $\mathbb{P}_\Lambda$ is the uniform probability measure on $D \times [v,w]^d \times [0,T]$. Thus the estimate in Theorem 4 can be viewed as an estimate in the space $\mathcal{L}^2(\mathbb{P}_\Lambda)$ and we want to emphasize that our setting easily allows us to choose arbitrary probability measures $\mathbb{P}$ on $D \times [v,w]^d \times [0,T]$ and prove analogous results w.r.t. the $\mathcal{L}^2(\mathbb{P})$ norm.

## 5 Conclusion

The method introduced in this paper is the first deep learning algorithm for the numerical solution of parametric Kolmogorov PDEs and one of few existing algorithms whose use is computationally tractable in high-dimensional settings. The parametric nature of our approach readily allows for sensitivity analysis, model calibration, and uncertainty quantification, all which is of high interest in a variety of applications. Successful numerical experiments in both low- and high-dimensional settings empirically confirm the functionality of the proposed algorithm. In addition, we are able to provide theoretical guarantees for the applicability of our method in high-dimensions.

Besides solving an important problem in scientific computing, our work introduces a class of learning problems that allows for the rigorous investigation of expressivity and sample complexity, along with stable and interpretable algorithms. Such strong results become possible by leveraging the mathematical structure of the learning problem associated with the parametric PDE. We anticipate that the formulation and study of such structured problems will constitute an important future direction of research in the scientific machine learning community as it can enable reliable and interpretable algorithms for the solution of previously intractable problems: in our case parametric families of Kolmogorov PDEs. This contributes substantially to areas like physical modelling of diffusion processes and computational finance, which all rely on the use of such PDEs.

## Broader Impact

The deep-learning technique presented in this work is the first computationally scalable method for the numerical solution of high-dimensional parametric Kolmogorov PDEs. It is also the first method which allows for a straightforward sensitivity analysis of the associated high-dimensional PDE solution manifold with respect to input parameters. In addition, it newly allows for high-dimensional data-driven model calibration and uncertainty quantification. While it is a difficult task to precisely estimate the cascading effects of technological innovations on wider society, it is reasonable to assume that the ubiquity of Kolmogorov equations in science and engineering will lead to a positive impact of our new findings on a multitude of technical areas of social importance.

As an example, Kolmogorov PDEs are heavily used in physics for the modelling of heat flow and diffusion processes [42, 58]. Simultaneously, Fokker-Planck equations, which take the form of Kolmogorov equations in particular special cases, are used in the geophysical and atmospheric sciences as modelling tools for climate change projections [25, 54]. Our described algorithm has clear promise to make previously intractable high-dimensional physical models computationally accessible to scientists. Additionally, our method allows for an easy investigation of changes in complex model forecasts as input parameters are varied during sensitivity analysis. Such advancements have the potential to accelerate scientific research and can directly lead to better predictive models in applied physics and engineering. Reliable and efficient predictive models in turn are essential to rationally inform public policy.

A conceivable risk posed by our work might come in the form of the uncritical use of our algorithm in applications related to financial engineering. The Black-Scholes equation and associated models have been notoriously misused in the last decades by semi-technical users working in financial sectors around the world [31, 59]. The naive usage of technical tools in computational finance has thus likely been a contributing factor to periods of economic instability in recent history. Our technique can now add a powerful solver for high-dimensional parametric PDE problems to the tool kits of individual end-users in finance with various degrees of scientific expertise. Inexperienced users without appropriate quantitative background might be prone to erroneously taking the complexity of a high-dimensional financial model as an indicator for its accuracy. Therefore, one must take great care to systematically inform users without suitable experience in such a scenario that merely increasing the dimension of an inadequate financial model might not necessarily make its results more accurate.

In total, we are confident that the net impact of our work on the scientific community as well as broader society is positive. The probability of uncritical use of our technique and other algorithms in financial engineering can likely be substantially mitigated by targeted educational interventions and we would encourage practical research in this direction. At the same time, we note that our technical contribution is a general-purpose tool which has the potential to stimulate the acceleration of scientific progress in a wide variety of disciplines.

## Acknowledgments and Disclosure of Funding

The research of Julius Berner was supported by the Austrian Science Fund (FWF) under grant I3403-N32. The research of Markus Dablander was supported by the UK EPSRC Centre For Doctoral Training in Industrially Focused Mathematical Modelling (EP/L015803/1).

## Footnotes

[2]Viscosity solutions are the appropriate solution concept for a wide range of PDEs [10, 23]. Viscosity solutions are continuous, but not necessarily differentiable.

[3]The classical way to circumvent the curse of dimensionality has been the employment of stochastic Monte Carlo based methods, see e.g. [19]; these techniques, however, are only suitable to approximately compute the solution $u_\gamma(x, t)$ at a single *fixed* space-time point $(x, t) \in [v, w]^d \times [0, T]$, limiting their usefulness in practice.

[4]We denote by $[a_1| \ldots |a_d] \in \mathbb{R}^{d \times d}$ the horizontal concatenation of the vectors $a_1, \ldots, a_d \in \mathbb{R}^d$.

[5]For the implementation details we refer the reader to Section A.2 in the appendix and the repository associated with this work on `https://github.com/juliusberner/deep_kolmogorov`.

[6]We know the underlying (uniform) distribution and therefore can normalize each input in an exact manner.

[7]Note that $\sigma_\gamma = \sigma_{\gamma,1}$ in the formal framework described in Section 2.1 but here and in the following we use the natural identifications, e.g. $D_\sigma \cong D_\sigma \times \{0\}$.

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
