[Supplementary Material 1]

# A Appendix

## A.1 Theoretical Results

First we state our conditions on the coefficient maps and initial conditions.

**Assumptions 1** (Coefficient Maps & Initial Conditions). *Let $D$ be a compact set in Euclidean space and for every $\gamma \in D$ let $\varphi_\gamma \in \mathcal{C}(\mathbb{R}^d, \mathbb{R})$, $\sigma_\gamma \in \mathcal{C}(\mathbb{R}^d, \mathbb{R}^{d \times d})$, and $\mu_\gamma \in \mathcal{C}(\mathbb{R}^d, \mathbb{R}^d)$. Assume that for every $x \in \mathbb{R}^d$ the mappings*

$$\gamma \mapsto \varphi_\gamma(x), \quad \gamma \mapsto \sigma_\gamma(x), \quad and \quad \gamma \mapsto \mu_\gamma(x)$$

*are continuous and that there exists $c \in (0, \infty)$ such that for every $\gamma \in D$, $x, y \in \mathbb{R}^d$ it holds that[8]*

(i) $|\varphi_\gamma(x) - \varphi_\gamma(y)| \le c\|x - y\|(1 + \|x\|^c + \|y\|^c)$,

(ii) $\|\mu_\gamma(x) - \mu_\gamma(y)\| + \|\sigma_\gamma(x) - \sigma_\gamma(y)\| \le c\|x - y\|$, *and*

(iii) $|\varphi_\gamma(0)| + \|\mu_\gamma(0)\| + \|\sigma_\gamma(0)\| \le c$.

Note that the continuity assumptions on $\sigma_\gamma$ and $\mu_\gamma$ and the condition in Item (ii) are fulfilled for the case of affine-linear coefficient functions as described in Section 2.1 and used in our examples. Further, the polynomial growth condition on the local Lipschitz constant in Item (i), the uniform bound in Item (iii), and the continuity assumption on $\varphi_\gamma$ are also satisfied for all our considered examples. Under these assumptions we can precisely formulate the setting we are working in.

**Setting** (Parametric Kolmogorov PDEs). *For every $\gamma \in D$ let $u_\gamma \colon \mathbb{R}^d \times [0, \infty) \to \mathbb{R}$ be the unique continuous, at most polynomially growing function satisfying for every $x \in \mathbb{R}^d$ that $u_\gamma(x, 0) = \varphi_\gamma(x)$ and satisfying that $u|_{\mathbb{R}^d \times (0,\infty)}$ is a viscosity solution of the Kolmogorov PDE*

$$\frac{\partial u_\gamma}{\partial t}(x, t) = \frac{1}{2} \operatorname{Trace} \left( \sigma_\gamma(x)[\sigma_\gamma(x)]^*(\nabla_x^2 u_\gamma)(x, t) \right) + \langle \mu_\gamma(x), (\nabla_x u_\gamma)(x, t) \rangle$$

*for $(x, t) \in \mathbb{R}^d \times (0, \infty)$, see [23, Corollary 4.17]. Let $(\Omega, \mathcal{F}, (\mathcal{F}_t)_{t \in [0, T]}, \mathbb{P})$ be a suitable filtered probability space satisfying the usual conditions, let*

$$(B_t)_{t \ge 0} \colon [0, \infty) \times \Omega \to \mathbb{R}^d \tag{11}$$

*be a standard d-dimensional $(\mathcal{F}_t)$-Brownian motion, let $T \in (0, \infty)$, $u \in \mathbb{R}$, $v \in (u, \infty)$ and let*

$$\Lambda = (\Gamma, X, \mathcal{T}) \colon \Omega \to D \times [v, w]^d \times [0, T]$$

*be a $\mathcal{F}_0$-measurable, uniformly distributed random variable. Let*

$$(S_{\gamma, x, t})_{t \ge 0} \colon [0, \infty) \times \Omega \to \mathbb{R}^d, \quad (\gamma, x) \in D \times [v, w]^d, \quad and \quad (S_{\Gamma, X, t})_{t \ge 0} \colon [0, \infty) \times \Omega \to \mathbb{R}^d$$

*be the up to indistinguishability unique $(\mathcal{F}_t)$-adapted stochastic processes with continuous sample paths satisfying that for every $(\gamma, x, t) \in D \times [v, w]^d \times [0, \infty)$ it holds $\mathbb{P}$-a.s. that*

$$S_{\gamma, x, t} = x + \int_0^t \mu_\gamma(S_{\gamma, x, s}) ds + \int_0^t \sigma_\gamma(S_{\gamma, x, s}) dB_s, \tag{12}$$

*and that for every $t \in [0, \infty)$ it holds $\mathbb{P}$-a.s. that*

$$S_{\Gamma, X, t} = X + \int_0^t \mu_\Gamma(S_{\Gamma, X, s}) ds + \int_0^t \sigma_\Gamma(S_{\Gamma, X, s}) dB_s, \tag{13}$$

*see, for instance, [17, Proof of Theorem 8.3]. For every $M \in \mathbb{N}$, $(\gamma, x, t) \in D \times [v, w]^d \times [0, \infty)$ let*

$$(S_{\gamma, x, t}^{M, m})_{m=0}^M \colon \{0, \ldots, M\} \times \Omega \mapsto \mathbb{R}^d$$

*be a stochastic process satisfying that $S_{\gamma, x, t}^{M, 0} = x$ and for every $m \in \{0, \ldots, M-1\}$ that*

$$S_{\gamma, x, t}^{M, m+1} = S_{\gamma, x, t}^{M, m} + \mu_\gamma(S_{\gamma, x, t}^{M, m}) \tfrac{t}{M} + \sigma_\gamma(S_{\gamma, x, t}^{M, m}) \left( B_{\frac{(m+1)t}{M}} - B_{\frac{mt}{M}} \right)$$

*and for every $M \in \mathbb{N}$ let*

$$(S^{M,m}_{\Gamma,X,\mathcal{T}})^M_{m=0} \colon \{0,\dots,M\} \times \Omega \mapsto \mathbb{R}^d$$

*be a stochastic process satisfying that $S^{M,0}_{\Gamma,X,\mathcal{T}} = X$ and for every $m \in \{0,\dots,M-1\}$ that*

$$S^{M,m+1}_{\Gamma,X,\mathcal{T}} = S^{M,m}_{\Gamma,X,\mathcal{T}} + \mu_\Gamma(S^{M,m}_{\Gamma,X,\mathcal{T}})\tfrac{\mathcal{T}}{M} + \sigma_\Gamma(S^{M,m}_{\Gamma,X,\mathcal{T}})\big(B_{\frac{(m+1)\mathcal{T}}{M}} - B_{\frac{m\mathcal{T}}{M}}\big).$$

*Finally, let the random variable $Y \colon \Omega \mapsto \mathbb{R}$ be given by*

$$Y := \varphi_\Gamma(S_\Lambda) = \varphi_\Gamma(S_{\Gamma,X,\mathcal{T}}),$$

*let*

$$(\Lambda_i, Y_i) \colon \Omega \mapsto \big(D \times [v,w]^d \times [0,T]\big) \times \mathbb{R}, \quad i \in \mathbb{N},$$

*be i.i.d. random variables with $(\Lambda_1, Y_1) \sim (\Lambda, Y)$, and for every $M \in \mathbb{N}$ let the random variable $Y^M \colon \Omega \mapsto \mathbb{R}$ be given by*

$$Y^M := \varphi_\Gamma(S^{M,M}_\Lambda) = \varphi_\Gamma(S^{M,M}_{\Gamma,X,\mathcal{T}}).$$

In order to prove Theorem 1 we assume the following regularity on our SDEs in (12) and (13).

**Assumptions 2** (Regularity Assumptions). *Assume that there exists a jointly measurable[9] function*

$$\Upsilon \colon \mathcal{C}([0,T], \mathbb{R}^d) \times D \times [v,w]^d \times [0,T] \to \mathbb{R}$$

*such that it holds $\mathbb{P}$-a.s. that*

$$\Upsilon(B, \Gamma, X, \mathcal{T}) = \varphi_\Gamma(S_\Lambda)$$

*and for every $(\gamma, x, t) \in D \times [v,w]^d \times [0,T]$ it holds $\mathbb{P}$-a.s. that*

$$\Upsilon(B, \gamma, x, t) = \varphi_\gamma(S_{\gamma,x,t}),$$

*where $B \colon \Omega \to \mathcal{C}([0,T], \mathbb{R}^d)$, $\omega \mapsto (t \mapsto B_t(\omega))$, denotes the mapping to the sample paths of the Brownian motion in (11).*

Note that the above assumptions are satisfied for the Black-Scholes model in Section 3.1 and the heat equations in Section 3.3. In the former case we can write

$$\Upsilon(b, \gamma, x, t) = \max\{\gamma_\varphi - xe^{-0.5t\,\gamma_\sigma^2 + \sqrt{t}\,\gamma_\sigma b(1)}, 0\}$$

and in the latter

$$\Upsilon(b, \gamma, x, t) = \|x + \sqrt{t}\,\gamma_\sigma b(1)\|^2 \quad \text{(paraboloid)}, \quad \Upsilon(b, \gamma, x, t) = e^{-\|x + \sqrt{t}\,\gamma_\sigma b(1)\|^2} \quad \text{(Gaussian)}$$

where $(b, \gamma, x, t) \in \mathcal{C}([0,T], \mathbb{R}^d) \times D \times [v,w]^d \times [0,T]$. Moreover, the existence of a suitable $\Upsilon$ is in general given for non-parametric Kolmogorov PDEs, see [17, Theorem 8.5] and [5]. First we establish that under our assumptions the minimizer of the statistical learning problem is indeed the parametric Kolmogorov PDE solution map.

**Theorem** (Learning Problem). *It holds that*

$$\bar{u} \colon D \times [v,w]^d \times [0,T] \to \mathbb{R}, \quad (\gamma, x, t) \mapsto \bar{u}(\gamma, x, t) := u_\gamma(x,t)$$

*is the (up to sets of Lebesgue measure zero) unique minimizer of the statistical learning problem*

$$\min_f \mathbb{E}\Big[\big(f(\Lambda) - Y\big)^2\Big] \tag{14}$$

*where the minimum is taken over all measurable functions $f \colon D \times [v,w]^d \times [0,T] \to \mathbb{R}$.*

*Proof.* Note that one can extend standard results on the moments of SDE solution processes (see [34, Theorems 4.5.3 and 4.5.4] and [16, Chapter 5, Theorem 2.3]) to prove that $S_\Lambda$ and thus also the target variable $Y = \varphi_\Gamma(S_\Lambda)$ have bounded moments. It is well-known that under this condition the (up to sets of measure zero w.r.t. the distribution of $\Lambda$) unique solution to the statistical learning problem (14) is given by the regression function

$$f^*(\gamma, x, t) := \mathbb{E}[Y \mid \Lambda = (\gamma, x, t)], \quad (\gamma, x, t) \in D \times [v,w]^d \times [0,T], \tag{15}$$

that is
$$f^* = \operatorname{argmin}_f \mathbb{E}\Big[\big(f(\Lambda) - Y\big)^2\Big],$$

see, for instance, [11]. Moreover, the Feynman-Kac formula establishes for every $(\gamma, x, t) \in D \times [v, w]^d \times [0, T]$ that
$$\mathbb{E}[\varphi_\gamma(S_{\gamma,x,t})] = u_\gamma(x, t) = \bar{u}(\gamma, x, t), \tag{16}$$

see [23, Corollary 4.17]. Finally, Assumptions 2 and the independence of $B$ and $\Lambda$ ensure that for every Borel measurable set $A \subseteq D \times [v, w]^d \times [0, T]$ it holds that

$$\begin{aligned}
\mathbb{E}\big[\mathbf{1}_{\{\Lambda \in A\}} \varphi_\Gamma(S_\Lambda)\big] &= \int_A \int_{\mathcal{C}([0,T], \mathbb{R}^d)} \Upsilon(b, \gamma, x, t) \, d\mathbb{P}_B(b) \, d\mathbb{P}_{(\Gamma, X, \mathcal{T})}(\gamma, x, t) \\
&= \int_A \mathbb{E}\big[\varphi_\gamma(S_{\gamma,x,t})\big] \, d\mathbb{P}_{(\Gamma, X, \mathcal{T})}(\gamma, x, t)
\end{aligned}$$

where we denote the distributions of $\Lambda$ and $B$ by $\mathbb{P}_{(\Gamma, X, \mathcal{T})}$ and $\mathbb{P}_B$ (Wiener measure), respectively. Together with the fact that $\Lambda$ is uniformly distributed, this proves that for almost every $(\gamma, x, t) \in D \times [v, w]^d \times [0, T]$ it holds that

$$\mathbb{E}[Y \,|\, \Lambda = (\gamma, x, t)] = \mathbb{E}[\varphi_\Gamma(S_\Lambda) \,|\, \Lambda = (\gamma, x, t)] = \mathbb{E}[\varphi_\gamma(S_{\gamma,x,t})],$$

see [46, Chapter 4] and [1, Theorem 13.46]. Combined with (15) and (16), this proves the claim. $\quad\square$

Next, we establish the stability of the previous result w.r.t. approximate data generation via the Euler-Maruyama scheme.

**Theorem** (Approximated Learning Problem). *For every $M \in \mathbb{N}$ let*
$$\bar{u}^M \colon D \times [v, w]^d \times [0, T] \to \mathbb{R}$$

*be the (up to sets of Lebesgue measure zero) unique solution to the approximated learning problem*

$$\min_f \mathbb{E}\Big[\big(f(\Lambda) - Y^M\big)^2\Big]$$

*where the minimum is taken over all measurable functions $f \colon D \times [v, w]^d \times [0, T] \to \mathbb{R}$. Then there exists a constant $C > 0$ such that for every $M \in \mathbb{N}$ it holds that*

$$\|\bar{u}^M - \bar{u}\|_{\mathcal{L}^\infty(D \times [v,w]^d \times [0,T])} \leq \frac{C}{\sqrt{M}}.$$

*Proof.* Extending results on the Euler-Maruyama scheme (see, e.g., [34, Theorem 10.2.2]) one can prove that also in the parametric case for every $p \geq 2$ there exists a constant $C > 0$ such that for every $M \in \mathbb{N}$, $(\gamma, x, t) \in D \times [v, w]^d \times [0, T]$ it holds that

$$\mathbb{E}\big[\|S_{\gamma,x,t}^{M,M}\|^p\big] \leq C \quad \text{and} \quad \big(\mathbb{E}\big[\|S_{\gamma,x,t}^{M,M} - S_{\gamma,x,t}\|^p\big]\big)^{1/p} \leq \frac{C}{\sqrt{M}}. \tag{17}$$

Similar to the previous proof one can further establish that for every $M \in \mathbb{N}$ and almost every $(\gamma, x, t) \in D \times [v, w]^d \times [0, T]$ it holds that

$$\bar{u}^M(\gamma, x, t) = \mathbb{E}[Y^M \,|\, \Lambda = (\gamma, x, t)] = \mathbb{E}[\varphi_\Gamma(S_\Lambda^{M,M}) \,|\, \Lambda = (\gamma, x, t)] = \mathbb{E}[\varphi_\gamma(S_{\gamma,x,t}^{M,M})]$$

where the existence of functions $\Upsilon^M$ with analogous properties as in Assumptions 2 are guaranteed by the Euler-Maruyama scheme. The local Lipschitz property of $\varphi_\gamma$ now ensures that for every $M \in \mathbb{N}$ and almost every $(\gamma, x, t) \in D \times [v, w]^d \times [0, T]$ it holds that

$$\begin{aligned}
|\bar{u}^M(\gamma, x, t) - \bar{u}(\gamma, x, t)| &= \big|\mathbb{E}\big[\varphi_\gamma(S_{\gamma,x,t}^{M,M})\big] - \mathbb{E}[\varphi_\gamma(S_{\gamma,x,t})]\big| \\
&\leq c \, \mathbb{E}\big[\|S_{\gamma,x,t}^{M,M} - S_{\gamma,x,t}\|\big(1 + \|S_{\gamma,x,t}^{M,M}\|^c + \|S_{\gamma,x,t}\|^c\big)\big]
\end{aligned} \tag{18}$$

which together with the Cauchy-Schwarz inequality and (17) proves the theorem. $\quad\square$

Note that this result can also be used to show that our generalization result in Theorem 4 is not compromised by using data simulated by the Euler-Maruyama scheme.

Now we outline how to prove the simultaneous approximation of the parametric solution map and its partial derivatives by a neural networks without curse of dimensionality, i.e. with the network size scaling only polynomially in the underlying spatial dimension. In mathematical terms, we prove approximation results in the Sobolev norm $\|\cdot\|_{W^{1,\infty}}$, see [15]. As a motivating example, we take the heat equation from Section 3.3 and from now on we only consider feed-forward neural networks with ReLU activation function (ReLU networks), see e.g. [44, Section 2] for a precise definition.

**Theorem** (Sobolev Approximation). *Let $a \in \mathbb{R}$, $b \in (a, \infty)$ and for every $d \in \mathbb{N}$ let*

$$\bar{u}_d(\gamma_\sigma, x, t) = \|x\|^2 + t \operatorname{Trace}(\gamma_\sigma \gamma_\sigma^*), \quad (\gamma_\sigma, x, t) \in [a,b]^{d \times d} \times \mathbb{R}^d \times [0, T],$$

*be the parametric solution map for the $d$-dimensional heat equation with paraboloid initial condition. Then there exists a constant $C > 0$ with the following property: For every $\varepsilon \in (0, 1/2)$, $d \in \mathbb{N}$ there exists a ReLU network $\Phi_{\varepsilon,d}$ with at most $\lfloor Cd^4 \log(d/\varepsilon) \rfloor$ parameters satisfying that*

$$\|\Phi_{\varepsilon,d} - \bar{u}_d\|_{W^{1,\infty}([a,b]^{d \times d} \times [v,w]^d \times [0,T])} \leq \varepsilon.$$

*Proof.* Our result is based on the following ReLU network approximation result in [22, Proposition C.1.], which is an extension of the work by Yarotsky [60]. Let $\Delta > 0$ and let $\operatorname{sq}: [-\Delta, \Delta] \to \mathbb{R}$ be the squaring function given by $\operatorname{sq}(x) := x^2$. Then there exists a ReLU network $\Phi_\varepsilon^{sq}$ with $\mathcal{O}(\log(1/\varepsilon))$ layers, $\mathcal{O}(1)$ neurons per layer, and parameters bounded by $\mathcal{O}(1)$ satisfying that

$$\|\Phi_\varepsilon^{sq} - \operatorname{sq}\|_{W^{1,\infty}([-\Delta,\Delta])} \leq \varepsilon.$$

By the polarization identity $xy = \frac{1}{2}((x+y)^2 - x^2 - y^2)$ an analogous result holds for the multiplication function $\operatorname{mult}: [-\Delta, \Delta]^2 \to \mathbb{R}$ given by $\operatorname{mult}(x,y) := xy$, see [22, Proposition C.2.]. We can therefore imitate the representation

$$\bar{u}_d(\gamma_\sigma, x, t) = \sum_{i=1}^d \operatorname{sq}(x_i) + \sum_{i,j=1}^d \operatorname{mult}\left(t, \operatorname{sq}((\gamma_\sigma)_{ij})\right)$$

using ReLU network concatenation and parallelization [14, Section 5]. Finally, we can estimate the error using a chain rule for ReLU networks [7]. □

Next, we show that our setting even allows for combined approximation and generalization results without curse of dimensionality. To prove this, we focus on the d-dimensional heat equation with varying diffusivity and Gaussian initial condition. We first show that ReLU networks are capable of efficiently approximating the parametric solution map.

**Theorem** (Approximation). *Let $a \in \mathbb{R}$, $b \in (a, \infty)$ and for every $d \in \mathbb{N}$ let*

$$\bar{u}_d(\gamma_\sigma, x, t) = \frac{1}{(1 + 2t\gamma_\sigma^2)^{d/2}} e^{-\frac{\|x\|^2}{1+2t\gamma_\sigma^2}}, \quad (\gamma_\sigma, x, t) \in [a,b] \times \mathbb{R}^d \times [0, T],$$

*be the parametric solution map of the $d$-dimensional heat equation with Gaussian initial condition. Then there exists a constant $C > 0$ with the following property: For every $\varepsilon \in (0, 1/2)$, $d \in \mathbb{N}$ there exists a ReLU network $\Phi_{\varepsilon,d}$ with at most $\lfloor C \operatorname{polylog}(d/\varepsilon) \rfloor$ layers, at most $\lfloor Cd \rfloor$ neurons per layer, and parameters bounded by $C$ satisfying that*

$$\|\Phi_{\varepsilon,d} - \bar{u}_d\|_{\mathcal{L}^\infty([a,b] \times [v,w]^d \times [0,T])} \leq \varepsilon.$$

*Proof.* The proof is based on combining ReLU approximation results for Chebyshev polynomials (see [21, Lemma III.6]), Gaussians (see [21, Theorem VIII.5]), and the squaring and multiplication functions $\operatorname{sq}, \operatorname{mult}$ (see the proof of the previous theorem). Specifically, for given $\Delta > 0$ we can approximate the function

$$[0, \Delta] \ni x \mapsto h(x) := \sqrt{\frac{1}{1+2x}}$$

up to precision $\varepsilon$ by ReLU networks with $\mathcal{O}(\operatorname{polylog}(1/\varepsilon))$ layers, $\mathcal{O}(1)$ neurons per layer, and parameters bounded by $\mathcal{O}(1)$, see [21, Lemma III.6][10]. Furthermore, the Gaussian

$$\mathbb{R}^d \ni x \mapsto g(x) := e^{-\|x\|^2} \tag{19}$$

can be globally approximated up to precision $\varepsilon$ by ReLU networks with $\mathcal{O}(\mathrm{polylog}(1/\varepsilon))$ layers, $\mathcal{O}(d)$ neurons per layer, and parameters bounded by $\mathcal{O}(1)$, see [21, Theorem VIII.5]. Finally, observe that

$$\bar{u}_d(\gamma_\sigma, x, t) = \mathrm{mult}\left(g\big((\mathrm{mult}(x_i, f(t, \gamma_\sigma)))_{i=1}^d\big), \mathrm{pow}_d\big(f(t, \gamma_\sigma)\big)\right)$$

where[11]

$$f(t, \gamma_\sigma) := h(\mathrm{mult}(t, \mathrm{sq}(\gamma_\sigma))) = \sqrt{\tfrac{1}{1+2t\gamma_\sigma^2}} \quad \text{and} \quad \mathrm{pow}_d(x) := (\mathrm{sq} \circ \mathrm{sq} \circ \cdots \circ \mathrm{sq})(x) = x^d.$$

We can imitate this representation using ReLU network concatenation and parallelization [14, Section 5] and estimate the error via the mean value theorem. $\qquad\square$

Now we show that the number of samples $s$ in (7), needed to learn the parametric solution map $\bar{u}$, does not suffer from the curse of dimensionality, either. To satisfy boundedness assumptions commonly used in statistical learning theory, we restrict ourself to clipped ReLU networks whose output is assumed to be bounded by 1. This can be achieved by composing each ReLU network with a simple clipping function, which itself can be represented as a small ReLU network [8, Section A.4]. Note that this incorporates our prior knowledge that the parametric solution map of the heat equation with Gaussian initial condition satisfies $\|\bar{u}_d\|_{\mathcal{L}^\infty} \leq 1$.

**Theorem** (Generalization). *Let $a \in \mathbb{R}$, $b \in (a, \infty)$ and for every $d \in \mathbb{N}$ let*

$$\bar{u}_d(\gamma_\sigma, x, t) = \frac{1}{(1 + 2t\gamma_\sigma^2)^{d/2}} e^{-\frac{\|x\|^2}{1+2t\gamma_\sigma^2}}, \quad (\gamma_\sigma, x, t) \in [a, b] \times \mathbb{R}^d \times [0, T],$$

*be the parametric solution map of the $d$-dimensional heat equation with Gaussian initial condition and let*

$$V_d := \mathrm{vol}([a, b] \times [v, w]^d \times [0, T]) = T(b - a)(w - v)^d.$$

*Then there exists a constant $C > 0$ with the following property: For every $\varepsilon, \rho \in (0, 1/2)$, $d, s \in \mathbb{N}$ with $s \geq C(d/\varepsilon)^2 \, \mathrm{polylog}(d/\varepsilon) \log(1/\rho)$, there exists a neural network architecture $\mathcal{A}_{\varepsilon,d}$ with at most $\lfloor C \, \mathrm{polylog}(d/\varepsilon) \rfloor$ layers and at most $\lfloor Cd \rfloor$ neurons per layer such that every measurable empirical risk minimizer*

$$\hat{\Phi}_{\varepsilon,d,s} \colon \Omega \to \mathcal{H}_{\varepsilon,d}, \quad \hat{\Phi}_{\varepsilon,d,s}(\omega) \in \arg\min_{\Phi \in \mathcal{H}} \frac{1}{s} \sum_{i=1}^s (\Phi(\Lambda_i(\omega)) - Y_i(\omega))^2, \quad \omega \in \Omega,$$

*over an hypothesis space $\mathcal{H}_{\varepsilon,d}$ of clipped ReLU networks with architecture $\mathcal{A}_{\varepsilon,d}$ and parameters bounded by $C$ satisfies that*

$$\mathbb{P}\left[\frac{1}{V_d} \|\hat{\Phi}_{\varepsilon,d,s} - \bar{u}_d\|_{\mathcal{L}^2([a,b] \times [v,w]^d \times [0,T])}^2 \leq \varepsilon\right] \geq 1 - \rho.$$

*Proof.* To simplify notation, we define $\|\cdot\|_{\mathcal{L}^2} := \|\cdot\|_{\mathcal{L}^2([a,b] \times [v,w]^d \times [0,T])}$ and for every $\Phi \in \mathcal{H}_{\varepsilon,d}$ we define its risk $\mathcal{R}(\Phi)$ and its empirical risk $\hat{\mathcal{R}}(\Phi)$ by

$$\mathcal{R}(\Phi) := \mathbb{E}\left[\big(\Phi(\Lambda) - \varphi_\Gamma(S_\Lambda)\big)^2\right] \quad \text{and} \quad \hat{\mathcal{R}}(\Phi) := \frac{1}{s} \sum_{i=1}^s (\Phi(\Lambda_i) - Y_i)^2.$$

The fact that the regression function coincides with the parametric solution map (see Theorem 1) and the bias-variance decomposition (see [8, 11]) imply that

$$\frac{1}{V_d} \|\hat{\Phi}_{\varepsilon,d,s} - \bar{u}_d\|_{\mathcal{L}^2}^2 = \underbrace{\mathcal{R}(\hat{\Phi}_{\varepsilon,d,s}) - \mathcal{R}(\Phi^*)}_{\text{generalization error}} + \underbrace{\frac{1}{V_d} \|\Phi^* - \bar{u}_d\|_{\mathcal{L}^2}^2}_{\text{approximation error}}$$

where $\Phi^* \in \arg\min_{\Phi \in \mathcal{H}_{\varepsilon,d}} \|\Phi - \bar{u}_d\|_{\mathcal{L}^2}$ is a best approximation of $\bar{u}_d$ in $\mathcal{H}_{\varepsilon,d}$. The previous theorem ensures that there exists a clipped ReLU network $\Phi_{\varepsilon,d} \in \mathcal{H}_{\varepsilon,d}$ satisfying that

$$\frac{1}{V_d} \|\Phi^* - \bar{u}_d\|_{\mathcal{L}^2}^2 \leq \frac{1}{V_d} \|\Phi_{\varepsilon,d} - \bar{u}_d\|_{\mathcal{L}^2}^2 \leq \|\Phi_{\varepsilon,d} - \bar{u}_d\|_{\mathcal{L}^\infty([a,b] \times [v,w]^d \times [0,T])}^2 \leq \varepsilon/2.$$

For the generalization error we make use of results on the covering numbers of neural network hypothesis spaces, see e.g. [8, Proposition 2.8]. They ensure the existence of clipped ReLU networks $(\Phi_i)_{i=1}^n \subset \mathcal{H}_{\varepsilon,d}$ with $\log(n) \in \mathcal{O}(d^2 \operatorname{polylog}(d/\varepsilon) \log(1/r))$ such that balls of radius $r$ (w.r.t. the uniform norm) around those functions cover $\mathcal{H}_{\varepsilon,d}$. We can then use the (uniform) Lipschitz continuity of the (empirical) risk to bound the generalization error by

$$\mathcal{R}(\hat{\Phi}_{\varepsilon,d,s}) - \mathcal{R}(\Phi^*) \leq \mathcal{R}(\hat{\Phi}_{\varepsilon,d,s}) - \hat{\mathcal{R}}(\hat{\Phi}_{\varepsilon,d,s}) + \hat{\mathcal{R}}(\Phi^*) - \mathcal{R}(\Phi^*)$$

$$\leq 2r\big[\operatorname{Lip}(\mathcal{R}) + \operatorname{Lip}(\hat{\mathcal{R}})\big] + 2\max_{i=1}^n \big|\mathcal{R}(\Phi_i) - \hat{\mathcal{R}}(\Phi_i)\big|.$$

Employing Hoeffding's inequality [28] and a union bound, it holds that

$$\mathbb{P}\big[\max_{i=1}^n \big|\mathcal{R}(\Phi_i) - \hat{\mathcal{R}}(\Phi_i)\big| \leq \varepsilon/8\big] \geq 1 - \rho.$$

where we need $s \in \mathcal{O}(\log(n/\rho)/\varepsilon^2)$ many samples. Thus, choosing $r \sim \varepsilon$ implies the claim. $\qquad\square$

## A.2  Implementation Details

First, we want to present a rigorous definition of our Multilevel network architecture.

**Definition 1** (Multilevel Architecture). *Let $L, q, p \in \mathbb{N}$, $\chi \in \{0,1\}$, and $\varrho\colon \mathbb{R} \to \mathbb{R}$. We define the Multilevel network $\Phi\colon \mathbb{R}^p \to \mathbb{R}$ with input dimension $\dim_{in}(\Phi) = p$, $L$ levels, amplifying factor $q$, (component-wise applied) activation function $\varrho$, and residual constant $\chi$ for every $x \in \mathbb{R}^p$ by*

$$\Phi(x) := \sum_{l=0}^{L-1} \Phi_l^{2^l}(x) \in \mathbb{R} \tag{20}$$

*where for every $l \in \{0, \ldots, L-1\}$, $i \in \{2, \ldots, 2^l\}$ the intermediate network outputs $\Phi_l^i(x)$ are given by*

$$\Phi_l^i(x) = \mathcal{A}_l^i(\varrho \operatorname{Norm}_l^i(\Phi_l^{i-1}(x) + \chi \Phi_{l+1}^{2i-2}(x)))$$

*and*

$$\Phi_l^1(x) = \mathcal{A}_l^1(\varrho(\operatorname{Norm}_l^1(\mathcal{A}_l^0(x)))) \quad \text{and} \quad \Phi_L^{2i}(x) = 0.$$

*In the above, the constant $\chi$ controls whether we use intermediate residual connections, and for every $l \in \{0, \ldots, L-1\}$ the functions*

$$\operatorname{Norm}_l^i\colon \mathbb{R}^{qp} \to \mathbb{R}^{qp}, \quad i \in \{1, \ldots, 2^l\},$$

*are denoting normalization layers, e.g. batch normalization [30] or layer normalization [3], and*

$$\mathcal{A}_l^0\colon \mathbb{R}^p \to \mathbb{R}^{qp}, \quad \mathcal{A}_i^l\colon \mathbb{R}^{qp} \to \mathbb{R}^{qp}, \quad i \in \{1, \ldots, 2^l - 1\}, \quad \mathcal{A}_l^{2^l}\colon \mathbb{R}^{qp} \to \mathbb{R}$$

*are learnable linear mappings (or affine-linear in case of $\mathcal{A}_l^{2^l}$).*

In the implementation of our examples we used $\chi = 1$ to propagate intermediate residuals from the corresponding higher level using additive skip-connections, followed by a Batch normalization layer as proposed by [30]. This allows the length of the shortest gradient path during backpropagation to scale like the number of levels $L$ instead of the number of layers $2^L$; a feature commonly known to prevent diminishing or exploding gradients [61]. Thus, we can maintain computational tractability while at the same time having rather deep architectures. Note that a certain depth is needed for our approximation and generalization results in Section A.1, as well as to optimally approximate certain families of functions [41, 44, 60]. We pick the ReLU activation function as non-linearity to remain consistent with our theoretical guarantees in Section A.1 and with the growing body of literature on the approximation and generalization capabilities of ReLU networks. To optimize the networks we use the Adam optimizer (with decoupled weight decay regularization as proposed by [40]) and exponentially decaying learning rate. The precise setup is summarized in Table 5 and the hyperparameters over which we optimized using Tune [38, 39] are given in Table 6.

Table 5: Training setup

| | Black-Scholes | Basket Put | Heat Paraboloid | Heat Gaussian |
|---|---|---|---|---|
| **Input sets** | | | | |
| $D_\sigma$ | $[0.1, 0.6] \times \{0\}$ | $([0.1, 0.6]^{3\times3})^4$ | $\{\vec{0}\} \times [0,1]^{10\times10}$ | $\{\vec{0}\} \times [0,0.1]I_{150}$ |
| $D_\mu$ | $\{\vec{0}\}$ | $[0.1, 0.6]^{3\times4}$ | $\{\vec{0}\}$ | $\{\vec{0}\}$ |
| $D_\varphi$ | $[10, 12]$ | $[10, 12]$ | $\{\}$ | $\{\}$ |
| $[v, w]$ | $[9, 10]$ | $[9, 10]$ | $[0.5, 1.5]$ | $[-0.1, 0.1]$ |
| $[0, T]$ | $[0, 1]$ | $[0, 1]$ | $[0, 1]$ | $[0, 1]$ |
| **Network** | | | | |
| $\dim_{in}(\Phi)$ | 4 | 53 | 111 | 152 |
| architecture | Multilevel | Multilevel | Multilevel | Multilevel |
| $(L, q, \chi)$ | (4,5,1) | (4,5,1) | (4,4,1) | (4,4,1) |
| activation $\varrho$ | ReLU | ReLU | ReLU | ReLU |
| Norm layer | Batch norm | Batch norm | Batch norm | Batch norm |
| #parameters | 5.4K | 0.8M | 2.4M | 4.5M |
| **Training** | | | | |
| solution SDE | analytic | Euler-Maruyama | analytic | analytic |
| optimizer | AdamW | AdamW | AdamW | AdamW |
| param. init. | $\mathcal{U}([-\xi, \xi])$ | $\mathcal{U}([-\xi, \xi])$ | $\mathcal{U}([-\xi, \xi])$ | $\mathcal{U}([-\xi, \xi])$ |
| weight decay | 0.01 | 0.01 | 0.01 | 0.01 |
| batch-size | $2^{16}$ | $2^{17}$ | $2^{17}$ | $2^{17}$ |
| (initial lr., decay) | $(10^{-2}, 0.25)$ | $(10^{-3}, 0.4)$ | $(10^{-3}, 0.4)$ | $(10^{-3}, 0.4)$ |
| patience | 4000 | 4000 | 4000 | 4000 |
| **Validation** | | | | |
| solution PDE | analytic | MC-approx. | analytic | analytic |
| batch-size | $2^{16}$ | $2^{17}$ | $2^{17}$ | $2^{17}$ |
| #eval. batches | 150 | 1 | 150 | 150 |
| **Execution** | | | | |
| seeds | 0,1,2,3 | 0,1,2,3 | 0,1,2,3 | 0,1,2,3 |
| #GPUs per trial | 2 (Tesla V100) | 4 (Tesla V100) | 2 (Tesla V100) | 2 (Tesla V100) |

1. **Input sets:** input sets for the parameter $\gamma$, the spatial variable $x$, and the time variable $t$, as defined in Section 2.1.

2. **Network:** input dimension $\dim_{in}(\Phi)$, activation function $\varrho$, number of levels $L$, amplifying factor $q$, usage of intermediate residual connections $\chi$, normalization layers $\text{Norm}$, and approximate number of network parameters as defined in Definition 1.

3. **Training:** solution method for the SDE, optimizer, initialization of the linear maps $\mathcal{A}_i^l$ where $\xi := d_{in}^{-1/2}$ with $d_{in}$ denoting the input dimension, weight decay, batch-size, initial learning rate, and factor for learning rate decay each patience steps. Note that the training data size in (7) is given by $s = \text{batch-size} \cdot \#\text{steps}$ where the number of steps is reported in our tables.

4. **Validation:** pointwise computation of the PDE solution, batch-size, and number of batches per evaluation.[12] Note that $n = \text{batch-size} \cdot \#\text{eval. batches}$ for each reported $\mathcal{L}^1$ error, see (9).

5. **Execution:** PyTorch module and random module seeds for the 4 independent runs, and number and type of GPUs per run.

Table 6: Ranges for hyperparameter optimization

| hyperparameter | range |
|---|---|
| $(L, q)$ | $\{3, 4\} \times \{4, 5, 6\}$ |
| optimizer | $\{$AdamW, SGD (with momentum & weight decay)$\}$ |
| batch-size | $\{16384, 32768, 65536, 131072\}$ |
| learning rate | $(10^{-1}, 10^{-5})$ |
| lr. decay factor | $(0.2, 0.6)$ |

Table 7: Ablation study for the Black-Scholes model

| | avg. time (s) | avg. best $\mathcal{L}^1$-error | #parameters |
|---|---|---|---|
| Feed-Forward + LayerNorm | $809 \pm 9$ | $0.1476 \pm 0.0772$ | 6741 |
| Feed-Forward + None | $496 \pm 26$ | $0.0526 \pm 0.0002$ | 6101 |
| Feed-Forward + BatchNorm | $3755 \pm 57$ | $0.0017 \pm 0.0003$ | 6741 |
| Multilevel $\chi = 0$ + LayerNorm | $867 \pm 10$ | $0.0349 \pm 0.0000$ | 5404 |
| Multilevel $\chi = 0$ + None | $570 \pm 6$ | $0.0069 \pm 0.0001$ | 4804 |
| Multilevel $\chi = 0$ + BatchNorm | $3414 \pm 18$ | $0.0012 \pm 0.0000$ | 5404 |
| Multilevel $\chi = 1$ + LayerNorm | $874 \pm 13$ | $0.0348 \pm 0.0001$ | 5404 |
| Multilevel $\chi = 1$ + None | $581 \pm 10$ | $0.0069 \pm 0.0000$ | 4804 |
| Multilevel $\chi = 1$ + BatchNorm | $3453 \pm 34$ | $\mathbf{0.0011} \pm 0.0001$ | 5404 |

Table 8: Ablation study for the heat equation with paraboloid initial condition

| | avg. time (s) | avg. best $\mathcal{L}^1$-error | #parameters |
|---|---|---|---|
| Feed-Forward | $14764 \pm 65$ | $0.0090 \pm 0.0003$ | 3020977 |
| Multilevel $\chi = 0$ | $13892 \pm 83$ | $0.0058 \pm 0.0001$ | 2380732 |
| Multilevel $\chi = 1$ | $14049 \pm 138$ | $\mathbf{0.0055} \pm 0.0001$ | 2380732 |

## A.3 Additional Numerical Results

In Tables 7 and 8 we present an ablation study which empirically proves the superior performance of our Multilevel architecture in combination with batch normalization compared to feed-forward architectures or the usage of layer normalization [3]. For the feed-forward architecture we used the network $\Phi_L^{2^L}$ defined in (20) (i.e. only the highest level of the corresponding Multilevel network with $L + 1$ layer and $\chi = 0$). Despite having slightly less parameters, our Multilevel architecture consistently outperforms the feed-forward architecture. Further, the use of residual connections, i.e. $\chi = 1$, has a positive impact. Note that all not-mentioned settings are kept as in Table 5.

The performance of our algorithm in the case of the Black-Scholes option pricing model from Section 3.1 is further illustrated in Figures 5, 6, 7, and 8. Finally, Figure 9 depicts the computational cost of our algorithm as a function of the problem input dimension for the heat equation with paraboloid initial condition.

Figure 5: Shows $\bar{u}(\gamma, x, \cdot)$ vs. the average prediction (and its standard deviation) at $x = 9.5$, $\gamma_\sigma = 0.35$, and $\gamma_\varphi = 11$.

Figure 6: Shows the Vega $\frac{\partial \bar{u}}{\partial \gamma_\sigma}(\gamma, x, \cdot)$ vs. the average prediction (and its standard deviation) at $x = 9.5$, $\gamma_\sigma = 0.35$, and $\gamma_\varphi = 11$.

Figure 7: Shows the average prediction error $\frac{|\Phi(\gamma,x,\cdot) - \bar{u}(\gamma,x,\cdot)|}{1 + |\bar{u}(\gamma,x,\cdot)|}$ and its standard deviation at $x = 9.5$, $\gamma_\sigma = 0.35$, and $\gamma_\varphi = 11$.

Figure 8: Shows the average error of the Vega $\frac{|\frac{\partial \Phi}{\partial \gamma_\sigma}(\gamma,x,\cdot) - \frac{\partial \bar{u}}{\partial \gamma_\sigma}(\gamma,x,\cdot)|}{1 + |\frac{\partial \bar{u}}{\partial \gamma_\sigma}(\gamma,x,\cdot)|}$ and its standard deviation at $x = 9.5$, $\gamma_\sigma = 0.35$, and $\gamma_\varphi = 11$.

Figure 9: Shows the cost in terms of number of network parameters times average number of steps to achieve an $\mathcal{L}^1$-error of $10^{-2}$ w.r.t. to the problem dimension $d^2 + d + 1$ for the heat equations with paraboloid initial condition and $d = 1, \ldots, 17$. The absence of the curse of dimensionality is underlined by the linear behaviour in the log-log inset. The error was evaluated every 250 steps and except of the varying dimension all settings are kept as in Table 5.

## Footnotes

[8]For a finite index set $I$ and $a, b \in \mathbb{R}^I$ we define $\|a\| = \sqrt{\sum_{i \in I} |a_i|^2}$ and $\langle a, b \rangle = \sum_{i \in I} a_i b_i$.

[9]If not further specified, we consider measurability w.r.t. the corresponding Borel sigma algebras.

[10]Note that we can choose uniformly bounded parameters by leveraging the depth of the network and the positive homogeneity of the ReLU activation function.

[11] If $d$ is not a power of 2 we make use of a hierarchical composition of multiplication and squaring functions, see also [14, Theorem 6.3].

[12]The evaluation of the PDE via Monte Carlo simulation as in (10) is computationally very expensive. That is the reason why we only took one evaluation batch per iteration for the Basket put option. However, note that training the network with Euler-Maruyama simulated data does not increase the training time significantly (see Table 2) which underlines the general applicability of our algorithm.


[Supplementary Material 2]



Legend:
- $y = 244415.6x^{2.36}$ (blue dashed)
- #parameters × avg. #steps (red solid with markers)
- ± std. (pink band)

x-axis: $\dim_{in}\Phi$