[Reviews · NeurIPS 2020]

Review 1

Summary and Contributions: The authors propose a neural network approach to solving Kolmogorov PDEs with parametrically varying coefficients via the Feynman-Kac formulation.

Strengths: The authors provide both empirical and theoretical support for their approach.

Weaknesses: The authors claim to circumvent the curse of dimensionality.

Correctness: Appears so.

Clarity: Sure.

Relation to Prior Work: Yes.

Reproducibility: Yes

Additional Feedback: Some comments: 1) In the Basket put option example, it is said that the SDE is simulated using Euler-Maruyama with 25 equidistant steps. However, since the time interval over which these 25 steps are taken is not presented it is impossible to judge how accurate the approximate samples are. 2) D is used to denote the domain of the \gamma parameter but also denotes some sort of operator (second equation in Thm. 2).


Review 2

Summary and Contributions: The paper proposes a new neural network based methodology for producing a global solution to a class of parameterised parabolic PDEs, as a function of space, time and the parameters. This is accomplished by producing training data from an SDE related to the PDE through the Feynman-Kac formula, and training a neural with a multilevel architecture designed to be amenable to the problem. The new approach escapes from the curse of dimensionality, in that the computational cost of the algorithm is sub-exponential in the parameter dimension. The authors test their new algorithm on a set of PDEs with affine linear parameters of increasing dimension to empirically demonstrate this sub-exponential dependence, and provide a set of theoretical guarantees on the accuracy of the global neural network approximation for a particular PDE. EDIT: I am pleased with the authors responses and happy with my original assessment.

Strengths: The paper is very strong; it is well written and presents an approach that addresses an existing problem in numerical analysis using state of the art techniques from machine learning. As such the contribution is clearly significant and novel, and relevant to the NeurIPS community. The theoretical results provided are sensible if narrow. The empirical evaluation is for a challenging set of test problems that are both realistic, in the sense that the both option pricing models remain highly relevant, and challenging in the sense that the parameter dimension is large, so that evaluating prices and derivatives for a large set of parameters would be computationally demanding.

Weaknesses: The authors restrict to affine-linear parameters, but this restriction is not a requirement of the methodology and it is unclear from the text why the restriction was put in place. Is the purpose solely to narrow the scope of the analysis, or is there some performance loss for more general coefficients? It would be useful for the authors to comment in the paper on why this restriction is imposed. Similarly the theoretical results proven apply only to the example from Section 3.3 with a specific initial condition. A more general result would be preferable, particularly since the highest dimension application the authors have provided is actually for a different initial condition (though, of course, the input dimension for the parabaloid initial condition is still rather high). However a more general result may not be straightforward to derive. In the numerical experiments, figure 3 is rather unclear - the lines are all so close to each other that one can only make out one line, and while this certainly suggests that the method is accurate it is impossible to determine the level of accuracy from such a plot. Plotting error scaled by standard deviation might be more informative. In addition, the number of input parameters is varied but not in a systematic way. It would be interesting to see plots of steps taken to achieve a given accuracy vs. dimension of parameter space, to more explicitly demonstrate the non-exponential cost of the method as a function of parameter dimension.

Correctness: All of the claims and methodology appear to be correct.

Clarity: The paper is excellently written, very clear and easy to follow.

Relation to Prior Work: The differences from other neural-network based methods for solving these problems, as well as classical numerical methods, are thoroughly discussed.

Reproducibility: Yes

Additional Feedback: I am unsure what the notation after line 134 means (is this supposed to be concatenating vectors into a matrix...?) Perhaps the authors could clarify? Perhaps I have missed this, but the scalar n in equation 7 is not explicitly mentioned; it would be good if this could be clarified.


Review 3

Summary and Contributions: The submission proposes a new architecture to learn approximate solutions to a parametric family of Kolmogorov PDEs. Theoretical results regarding the ability of ReLU nets to approximate the solution and generalization bounds for the ERM are presented.

Strengths: The paper is clearly written. The theoretical derivations are clean.

Weaknesses: There are two main weaknesses of this work in my opinion. The first is regarding the generation of the ground-truth data, where Feynman–Kac is invoked and an SDE is simulated. While for sections 3.1 and 3.3, the associated SDEs have analytical solutions and can thus be solved precisely, it is unclear how the quality of the data should affect the learned solution in section 3.2. I think the discretization step size and number of samples used to approximate the F-K formula's expectation should both matter for the learning algorithm. It would be useful to conduct ablation studies to see the sensitivity of the algorithm to the data quality. Additionally, the theoretical results seem somewhat disconnected to the empirically driven algorithm, since the theorems ignore the concern of whether gradient-descent-like algorithm such as Adam and SGD can actually find the ERM. Understandably, this is a hard problem. Though, I believe it's imprecise to say that "... and rigorously show that our algorithm does not suffer from the curse of dimensionality" as on line 38. The curse of dimensionality in the PDE context usually refers to the exponential growth in the number of computational operations, and this is not directly comparable to the statements that ERM neural nets have number of parameters polylog in d, since it is definitely not obvious that the ERM can actually be found with time complexity polylog in d.

Correctness: The methodology and theoretical proofs are correct, to the best of my knowledge.

Clarity: The paper is well-written in my opinion.

Relation to Prior Work: The relation is discussed adequately in my opinion. Though, it is possible that I am unaware of some works that use NNs to learn PDEs, since I don't actively work in this area.

Reproducibility: Yes

Additional Feedback: Minor - line 171: "an European" -> "a European" - appendix 464: "in order to proof" -> "in order to prove" Post-rebuttal: I appreciate the authors in attempting to address my concerns, and the response has mitigated my concern regarding COD. I'm therefore raising my score to 7. With regards to data quality vs accuracy of PDE sol'n, my original intention was not to demand a theorem proving convergence (without explicit constants), although it's not bad that we have this guarantee now. In some sense I view this paper as a numerics methodology paper, and towards justifying that this is a useful method to practitioners, it would be useful to showcase some experimental results (on at least toy examples) that study the solution quality's dependence on some of the parameters (e.g. time horizon, volume in Euclidean space,...). These dependences are typically hard to obtain by deriving bounds. Moreover, the obtained bounds are hard to make tight (or prove tight via lower bound arguments). Overall, I think while the paper's scope is somewhat narrow, it's doing a great job in making statements precise and clean.

[Author Response · NeurIPS 2020]

We thank the reviewers for their useful feedback. We will carefully address all raised questions in the camera-ready version. Additionally, we will incorporate detailed new material based on the clarifications outlined below. This will go along with a treatment of the reviewers' smaller improvement suggestions on notation and figure design. Further, we intend to increase the number of facilitating pointers to the appendix, where detailed theorem formulations and supplementary information on numerical quantities such as time interval parameters can be found.

**(Non-) Necessity of Affine-Linear Coefficients (reviewer #3):** In our numerical examples, we focus on affine-linear coefficient functions because they are important in practical applications, computationally fast to evaluate, and easy to parametrize. As rightfully pointed out by one reviewer, however, the presented method is indeed not restricted to the case of affine-linear coefficients and can as well be used in a substantially more general setting. In particular, note that affine-linear coefficients are not assumed when the rigorous validity of the core learning problem is established in Theorem 1.

**Robustness w.r.t. Approximate Data Generation via Euler-Maruyama (reviewers #2, #5) :** Let $S_\Lambda^N$ be the Euler-Maruyama approximation of the solution to the parametric SDE $S_\Lambda = S_{\Gamma,X,\mathcal{T}}$ (as defined in (10) in the paper) with $N \in \mathbb{N}$ equidistant steps, given by

$$S_\Lambda^0 = X \quad \text{and} \quad S_\Lambda^{n+1} = S_\Lambda^n + \mu_\Gamma(S_\Lambda^n)\frac{\mathcal{T}}{N} + \sigma_\Gamma(S_\Lambda^n)\big(B_{\frac{(n+1)\mathcal{T}}{N}} - B_{\frac{n\mathcal{T}}{N}}\big), \quad n = 0, \ldots, N-1.$$

We managed to prove a theorem which shows that using our method with data obtained via the Euler-Maruyama scheme must result in the expected approximation of the parametric PDE solution map $\bar{u}$.

**Theorem.** *Assume Assumptions* 1 *and* 2 *from the appendix and further assume that $\varphi_\gamma$ has an at most polynomially growing derivative. Let $\bar{u}$ be the parametric solution map of the Kolmogorov PDE. Then there exists a constant $C$ depending only on $v, w, T$, and the growth rates and (local) Lipschitz constants of $\sigma_\gamma$, $\mu_\gamma$, and $\varphi_\gamma$ such that the solution to the approximated learning problem $\bar{u}^N = \arg\min_f \mathbb{E}\big[\big(f(\Lambda) - \varphi_\Gamma(S_\Lambda^N)\big)^2\big]$ satisfies that*

$$\max_{(\gamma,x,t)\in D\times[v,w]^d\times[0,T]} |\bar{u}^N - \bar{u}| \leq \frac{C}{\sqrt{N}}.$$

*Proof (Sketch).* Extending results on the Euler-Maruyama scheme (see, e.g., [Kloeden and Platen, 1992, Theorem 10.2.2]) one can prove that also in the parametric SDE case for $p \geq 2$ the $p$-th moments of $S_\Lambda$ and $S_\Lambda^N$ are bounded and that it holds that $\big(\mathbb{E}\big[\|S_\Lambda - S_\Lambda^N\|_{\mathbb{R}^d}^p\big]\big)^{1/p} \leq \frac{C}{\sqrt{N}}$. The local Lipschitz property of $\varphi_\gamma$ then proves the claim. $\square$

This result provides a theoretical guarantee for the robustness of our machine learning method; it can easily be used to prove that our generalization results are not compromised by using data simulated by the Euler-Maruyama scheme.

**The factor $1/V$ (reviewer #2):** The factor $\frac{1}{V} = \frac{1}{\text{vol}(D\times[v,w]^d\times[0,T])}$ naturally appears when transforming $L^\infty$- to $L^p$-results and can be omitted by viewing the error in the space $L^p(\mathbb{P}_\Lambda)$ (where $\mathbb{P}_\Lambda$ is the uniform probability measure on $V$) via

$$\|\cdot\|_{L^p(\mathbb{P}_\Lambda)}^p = \frac{1}{V}\|\cdot\|_{L^p(D\times[v,w]^d\times[0,T])}^p \leq \|\cdot\|_{L^\infty(D\times[v,w]^d\times[0,T])}^p.$$

All results within the established standard setting of statistical learning theory (including our generalization bound) give rise to $L^p$-bounds w.r.t. a given probability measure on the input domain. In fact, note that our setting easily allows us to choose arbitrary probability measures $\mathbb{P}$ on $D \times [v,w]^d \times [0,T]$ and prove analogous results w.r.t. the $L^2(\mathbb{P})$ norm. Thus, following the conventional terminology used in statistical learning, we can indeed claim that the presented bound overcomes the curse of dimensionality. To underline this we mention Barron [1993] as one of many examples of a classical and well-known approximation result where the terminology "avoiding/overcoming the curse of dimensionality" is used in strictly the same context as in our paper. We aim to further clarify this in the camera-ready version.

**Overcoming the Curse of Dimensionality (reviewer #5):** Based on the feedback of reviewer #5 we will further clarify in the camera-ready version that the curse of dimensionality is overcome with respect to the neural network size as well as the sample size. We emphasize that our empirical results strongly suggest that also the ERM algorithm does not suffer from the curse of dimensionality but proving this is out of scope of this paper.

# References

A. R. Barron. Universal approximation bounds for superpositions of a sigmoidal function. *IEEE Transactions on Information theory*, 39(3):930–945, 1993.

P. E. Kloeden and E. Platen. *Numerical solution of stochastic differential equations*, volume 23 of *Applications of Mathematics (New York)*. Springer-Verlag, Berlin, 1992. ISBN 3-540-54062-8.


[Meta-Review · NeurIPS 2020]

The three reviewers, who hail from different sub-communities that all overlap with the paper's content, agree that this is a very well presented work that combines rarely used techniques (such as Feynman-Kac) to interesting ML use cases. It should thus be accepted. The reviewers also raised some concerns about the presentation of the experiments. Please make sure to address these for the camera-ready version.